# Longitudinal Modulation of Loco-Regional Immunity in Ovarian Cancer Patients Receiving Intraperitoneal Chemotherapy

**DOI:** 10.3390/cancers14225647

**Published:** 2022-11-17

**Authors:** Adria Suarez Mora, Mary Strange, Yusi Fang, Ibrahim Uygun, Lixin Zhang, George C. Tseng, Pawel Kalinski, Robert P. Edwards, Anda M. Vlad

**Affiliations:** 1Department of Obstetrics and Gynecology and Reproductive Sciences, School of Medicine, University of Pittsburgh, Pittsburgh, PA 15213, USA; 2Magee-Womens Research Institute, Pittsburgh, PA 15213, USA; 3Magee-Womens Hospital of UPMC, Pittsburgh, PA 15213, USA; 4Department of Biostatistics, Graduate School of Public Health, University of Pittsburgh, Pittsburgh, PA 15260, USA; 5Roswell Park Comprehensive Cancer Center, Buffalo, NY 14203, USA

**Keywords:** ovarian cancer, chemotherapy, loco-regional immunity, Th2 immunity, complement system

## Abstract

**Simple Summary:**

Insight into how immune cells change during chemotherapy can help develop new ways to support immune function and maximize treatment success in epithelial ovarian cancer (EOC). We aim to define the impact of chemotherapy on the tumor immune microenvironment by studying changes in immune-related gene expression and proteins within fluid that surrounds EOC. By testing different time points during chemotherapy, we can gain insight into the interplay between immune cells and cancer cells. Longitudinal profiling of cellular and molecular changes in immune surveillance during treatment of ovarian cancer will help find more effective combination treatments that can improve outcomes in EOC.

**Abstract:**

The immune tumor microenvironment (TME) of epithelial ovarian cancer (EOC) carries both effector and suppressive functions. To define immune correlates of chemotherapy-induced tumor involution, we performed longitudinal evaluation of biomarker expression on serial biological specimens collected during intraperitoneal (IP) platinum-based chemotherapy. Serial biological samples were collected at several time points during IP chemotherapy. RNA from IP fluid cells and tumor tissue was analyzed via NanoString. Meso Scale Discovery (MSD) multiplex assay and ELISA for MUC1 antibodies were performed on plasma and IP fluid. Differentially expressed genes in IP fluid demonstrate an upregulation of B cell function and activation of Th2 immune response along with dampening of Th1 immunity during chemotherapy. MSD analysis of IP fluid and gene expression analysis of tumor tissue revealed activation of Th2 immunity and the complement system. Anti-MUC1 antibodies were detected in IP fluid samples. IP fluid analysis in a secondary cohort also identified chemotherapy-induced B cell function genes. This study shows that serial IP fluid sampling is an effective method to capture changes in the immune TME during chemotherapy and reveals treatment induced changes in B cell function and Th2 immunity.

## 1. Introduction

Epithelial ovarian cancer (EOC) is the fifth leading cause of cancer death in women and remains the most lethal gynecologic malignancy with 19,880 estimated cases and 12,810 estimated deaths in the US, in 2022 [1]. More than 75% of cases are diagnosed at advanced stages and have a high rate of recurrence [2]. Despite advances with upfront cytotoxic chemotherapy, which triggers clinical responses in 80% of patients, prognosis remains poor due to the lack of effective subsequent treatment options. Accumulating evidence demonstrates that the immune system plays an active role in ovarian cancer biology. For example, CD3+ tumor infiltrating lymphocytes (TILs) are present in over half of EOC tumor samples and correlate with improved survival [3,4]. The absence of CD8+ cytotoxic T lymphocytes (CTL) is also an independent predictor of platinum resistance [5]. However, the immune system can have suppressive functions through regulatory T cells (Tregs), myeloid derived suppressor cells and engagement of inhibitory immune checkpoint receptors (like PD-1/PD-L1, for example) [6,7].

In recent years, immunotherapy has evolved into an effective treatment modality because of the therapeutic benefit of immune checkpoint blockade seen in solid tumors like melanoma, lung, and urothelial cancers [8,9,10]. Nevertheless, the efficacy of single agent immunotherapy remains modest in EOC, pointing to the need for combination therapies [11]. Therefore, deeper understanding of the dynamic changes of the immune microenvironment during chemotherapy-induced tumor involution, including changes in its multiple component cells, may allow for more successful use of immunotherapy or combination chemo-immune therapy in EOC.

Regional therapy including intraperitoneal (IP) chemotherapy delivers cytotoxic drug directly to the tumor microenvironment (TME) in the peritoneal cavity. This is achieved using an IP catheter, a device implanted through the skin and tunneled into the peritoneal cavity. In addition to delivery of chemotherapy, the IP catheter provides a unique mechanism to sample the TME during treatment, via serial collection of IP fluid. We previously demonstrated that biomarker analysis is feasible using serial IP fluid [12]. Importantly, biomarker expression in the IP fluid differs from peripheral blood, offering a more accurate assessment of treatment-induced tumor involution [12,13,14].

We hypothesize that through sequential sampling of IP fluid during IP chemotherapy we can capture treatment-induced changes in the TME and better define the biology of tumor involution in EOC. Using biological specimens obtained from patients treated with IP chemotherapy, we profiled loco-regional inflammation at different time points during treatment revealing contrasting changes in loco-regional T and B cell immunity.

## 2. Materials and Methods

### 2.1. Patient Cohort and Sample Collection

These studies were conducted according to a protocol approved by the Institutional Review Board (IRB) of the University of Pittsburgh. Written informed consent was obtained from each patient for specimen banking. Additional IRB approval was obtained for the use of biological specimens and access to clinical information. Patients with EOC who received platinum-based IP chemotherapy, (IV paclitaxel 135 mg/m^2^, IP cisplatin 75 mg/m^2^, IP paclitaxel 60 mg/m^2^) and who had previously had serial biological specimens collected during chemotherapy were identified. Two patients received either carboplatin or oxaliplatin instead of cisplatin. Biological samples including blood and IP fluid were collected with IP catheter placement, prior to each cycle of IP chemotherapy, and with IP catheter removal. A schematic of specimen collection is shown in Figure 1.

Tumor tissue was obtained with placement and removal of the IP catheter. Due to logistics, some patients either did not have samples collected at every time point or volume was too low for processing. Clinical information was collected retrospectively through an honest broker.

Samples were processed within 24 h, most often within 4 h of specimen collection. Peritoneal sampling consisted of either peritoneal fluid or peritoneal wash, depending on what was clinically feasible, and are both referred to as IP fluid moving forward. Peritoneal fluid is fluid retrieved spontaneously from the peritoneal cavity and peritoneal wash is fluid obtained by flushing the IP catheter with 30 mL of sterile saline solution, and subsequently withdrawing the fluid. Samples were centrifuged to separate cells from supernatant. Cells were cryopreserved in 10% dimethyl sulfoxide (DMSO; Sigma, Kanagawa, Japan) and fetal bovine serum (FBS; R&D Systems, Minneapolis, MN, USA). Supernatant fluid was stored at −80 °C. Time points were defined as early (before treatment initiation), intermediate (after 2–4 cycles) and late (at completion).

Plasma and peripheral blood mononuclear cells (PBMCs) were also collected at early and late time points, as defined above. Plasma was stored at −80 °C. PBMCs were isolated using mononuclear cell preparation tubes (BD Biosciences, Franklin Lakes, NJ, USA) and cryopreserved in 10% DMSO/FBS freezing media. Formalin fixed paraffin embedded (FFPE) tissue sections from placement and removal of the IP catheter were obtained from the University of Pittsburgh Biospecimen Core.

A secondary cohort of patients was established from a completed Phase I/II clinical trial (NCT02432378) at our institution investigating the safety and efficacy of IP cisplatin with novel chemokine modulation including IP rintatolimod and IP interferon alpha (IFN⍺) [15]. Inclusion criteria were recurrent epithelial ovarian adenocarcinoma, platinum sensitivity (>6 months interval from previous platinum therapy) and measurable recurrent disease. The clinical trial was approved by the IRB of the University of Pittsburgh. Our analyses utilized IP wash samples from baseline and after one cycle of IP cisplatin (50 mg/m^2^) prior to administration of IP chemokine modulation. Samples were collected and processed as above.

### 2.2. NanoString

RNA was isolated from IP cells obtained from IP fluid using the AllPrep DNA/RNA/Protein Kit (Qiagen, Hilden, Germany). RNA from peripheral blood mononuclear cells (PBMCs) was isolated using the Qiagen AllPrep DNA/RNA Kit. RNA was also isolated from tumor tissue using the Qiagen AllPrep DNA/RNA FFPE Kit. Tissue was scrapped from 10 µm thick slices of FFPE tumor specimens using 2–5 slides per specimen. All RNA isolation was performed according to the manufacturer’s protocol. RNA concentration was measured with a NanoDrop 2000 UV-Vis Spectrophotometer (Thermo Scientific, Waltham, MA, USA).

NanoString was performed at the University of Pittsburgh Genomics Research Core on all RNA samples according to the standard protocol provided (nCounter XT CodeSet gene Expression Assays). The nCounter Human PanCancer Immune Profiling Panel kit was used for each analysis which contains probes for 730 immune genes plus 40 internal reference genes. 6 samples were concentrated using a SpeedVac Vacuum Concentrator to obtain adequate volumes for NanoString analysis. 50–100 ng of RNA were used for each sample. Hybridization reactions were incubated for 16 h. An nCounter MAX/FLEX was used to run the hybridization reactions on the nCounter platform using the high sensitivity protocol.

### 2.3. ELISA

Mucin -1 (MUC1) antibodies were measured by ELISA, using our previously published protocols [16]. MUC1 100 mer peptide was used (provided by Dr. Olivera Finn’s laboratory at the University of Pittsburgh School of Medicine, Department of Immunology) (1 µg/50 µL). Plasma was diluted 1:40 and IP fluid supernatant diluted 1:5. Secondary antibody was goat anti-human IgM or IgG (Sigma, Kanagawa, Japan) diluted 1:2000. Absorbance was read at 405 nm on the BioRad iMark Microplate Reader using BioRad Microplate Manager Software Version 6.1. IP fluid samples were run in duplicate and plasma samples in triplicate. Each sample was incubated in either MUC1 peptide (experimental) or albumin (PBS, background) coated wells and the absorbance was calculated as the difference between the experimental and background absorbance for each sample. A positive control, a patient sample known to have high antibody levels, was also included in each ELISA. A positive antibody level was defined as an absorbance value above the average background absorbance plus 3 standard deviations of background absorbance.

### 2.4. Immunohistochemistry (IHC)

IHC analysis was performed on 4 µm thick slices of FFPE tumor specimens as previously described [15]. 7 patients had sufficient paired pre-treatment and post-treatment tumor tissue for transcriptomic analysis. Primary antibodies were used at—1:100 for CD8 (Agilent, clone C8/144B, Santa Clara, CA, USA), 1:200 for CD20 (Abcam, clone EP459Y, Cambridge, UK), and 1:100 for CD3 (Agilent, Clone F7.2.38, Santa Clara, CA, USA). A matched isotype was also applied on a replicate slide as a negative control. Corresponding secondary antibody were Envision+ System HRP anti-mouse IgG (Agilent, Santa Clara, CA, USA) for CD3, CD8 and anti-rabbit IgG (Agilent, Santa Clara, CA, USA) for CD20. For antigen retrieval, slides were heated for 20 min in Tris-EDTA-Tween buffer pH 9 for all antibodies. Scoring for CD8+ T cells was performed according to the Ovarian Tumor Tissue Analysis Consortium protocol [17] using a Zeiss Axiostar Plus microscope and Zen imaging software. A 4-point ordinal score was used based on CD8+ T cell counts per high powered field: negative (none), low (1–2), moderate (3–19), and high (≥20). The maximum score was used for each slide.

### 2.5. Meso Scale Discovery (MSD)

IP fluid and plasma samples were analyzed with the MSD platform, according to the manufacturer’s protocols for U-PLEX Biomarker Group 1 Multiplex Assays and R-PLEX Singleplex Assays. The U-PLEX panel included IL-4, IL-5, IL-6, IL-8, IL-10, IL-17A, CXCL5 (ENA-78), CXCL10 (IP-10), CCL19 (MIP-3β), and CCL24 (Eotaxin-2). The R-PLEX assays consisted of perforin and MIG. The plates were run on the MSD MESO QuickPlex SQ 120 imager. Total protein in IP fluid and plasma samples was measured using the Pierce BCA Protein Assay Kit (Thermo Scientific, Waltham, MA, USA).

### 2.6. Statistics and Bioinformatics

Patient demographics were summarized using descriptive statistics. Gene expression data was normalized to the 20 housekeeping genes using the NanoStringNorm R package. Differentially expressed (DE) genes were identified with R package edgeR using likelihood ratio tests on negative binomial models for each gene [18]. Significance was also determined using the Benjamini-Hochberg procedure to calculate the *q*-value, which controls for the false discovery rate from multiple comparisons. *q*-values < 0.2 were set as cut-offs for exploratory analysis. When multiple comparisons were not corrected, *p*-values < 0.05 were considered significant. In order to have sufficient DE genes and statistical power to perform pathway enrichment analysis and understand the functional annotation in the detected DE genes, a *q*-value threshold (*q* < 0.2) was used when >30 genes were detected, otherwise a *p*-value threshold was used (*p* < 0.05). Unpaired analyses were performed to preserve statistical power of the comparisons of early versus intermediate time points and intermediate versus late time points. For consistency, unpaired DE gene analysis was used for all comparisons. Overlapping genes were visualized using Venn diagrams (Venny 2.1) [19]. The Ingenuity^®^ Pathway Analysis (IPA^®^) (QIAGEN Inc., Germantown, MD, USA, https://www.qiagenbioinformatics.com/products/ingenuitypathway-analysis, accessed on 4 October 2020) was used to identify enriched pathways in DE genes. Heatmaps were generated based on DE genes. For the MSD data, logarithm transformation was applied for the normalization prior to the identification of DE proteins. Protein concentration was also normalized based on the total protein content. Significantly changed proteins (*p*-values < 0.05) were identified using the paired-sample *t*-tests and ANOVA. Relative abundance ratios were calculated by comparing normalized protein concentrations in the IP fluid and plasma.

## 3. Results

### 3.1. Immune Gene Expression Analyses Capture Chemotherapy-Induced Dynamic Changes in IP Fluid

We collected serial biological samples from a cohort of 9 patients diagnosed with EOC who underwent treatment with IP platinum-based chemotherapy. Patient characteristics are summarized in Table 1. Most patients had stage III disease, serous histology, and were platinum sensitive on first- or second-line treatment.

8 patients had sufficient RNA from IP fluid for analysis, 1 patient was excluded due to insufficient RNA after baseline collection. All patients had samples at early (baseline) and late (cycle 5 to completion) time points; 4 patients had sufficient RNA from intermediate (cycle 2–4) time points. A total of 161 immune genes were differentially expressed (DE) between early and late time points via NanoString analysis (Figure 2A and Appendix A). The majority (90%) of these genes are upregulated suggesting an increase of immune function at treatment completion. DE genes between early-intermediate and intermediate-late time points follow a similar pattern with most being upregulated (54–62%; Figure 2B,C, Appendix A). Shared DE genes between early-intermediate and early-late time points identified eleven genes (Figure 2D, Appendix A). 6 out of these 11 genes (*TNFRSF13C, CXCR4, BLK, AICDA, CD19, CD79B*) are specific to B cell function and tertiary lymphoid structure (TLS) formation with most showing upregulation early during chemotherapy (Figure 2E). IPA^®^ on DE genes between early-intermediate time points identified 5 B cell signaling pathways in the top 10 canonical pathways, 2 of which are also identified between early-late and intermediate-late time points (Appendix A). This suggests B cells-specific gene expression in platinum-induced tumor involution is activated early in chemotherapy but may remain active even late in chemotherapy.

Shared DE genes between intermediate-late and early-late time points identified an additional 56 genes with the majority showing upregulation (88%, Appendix A). Closely tied to B cell function, Th2 genes were identified in the shared DE genes, including *IL-4*, *IL-25* and *IL-5*, which show upregulation (Figure 2E), pointing to activation of Th2 immune pathways. In contrast, among the 11 downregulated genes from early to late time points, 7 were specific to Th1 immunity and included *IL32*, *GZM*A, *GZM*B, *CD247*, *CD3E*, *IL1B* and *IL8* (Figure 2F).

We further profiled protein expression in the immune TME using measurements of chemokine and cytokine concentrations in the IP fluid samples. Using the MSD platform, we identified a trend for increased median concentration of IL4, IL5 and IL6 in IP fluid at intermediate and/or late time points, compared to baseline (Figure 2G). MSD data also shows a high enrichment of proteins (50- to 100-fold) of IL4, IL5 and IL6 in the peritoneal cavity, compared to peripheral blood, as reflected by the high ratio of IP wash-to-blood concentration of these cytokines (Figure 2H). Together, these results suggest a Th2 dominant loco-regional TME during platinum-based chemotherapy.

### 3.2. Tumor and IP Fluid Show Concordant Gene Expression Profiles of Th2 Genes

NanoString analysis of paired tissue samples from 7 patients revealed 68 genes that were DE between pre- and post-treatment time points (Appendix A). 66% were upregulated (Figure 3A) mirroring the trend seen in IP fluid. Comparison of DE genes from pre/post-treatment tumor tissues and early/late IP fluid resulted in 26 shared genes (Figure 3B, Appendix A) with 21 showing concordant directionality and enrichment of upregulated cytokines implicated in the Th2/cell-antibody immune response (*IL4, IL5,* and *IL25*—Figure 3C). Additionally, we noted that several genes related to the complement system (including *MASP1*, *C4BPA, MBL2, and C8A*), were also upregulated (Figure 3C). The complement system plays an important function in cytotoxicity against antibody coated target cells and may be closely related to the observed upregulation of Th2 (humoral immunity promoting responses) and B cell function in IP fluid cells and tumor tissue, as demonstrated by IPA^®^ in tumor tissue and the comparisons of IP fluid at three time points (Appendix A).

### 3.3. Tumor Infiltrating Immune Cell Populations Show Dampening of CD8+ T Cells with Chemotherapy Treatment

IHC was performed on pre- and post-treatment tumor tissue to identify and quantify CTLs (CD8), B cells (CD20) and T cells (CD3). Pre-treatment tissue samples showed high CD8+ T cells, with only 2 patients having low CD8+ T cells. Post-treatment, CD8+ T cell distribution changed with 4 patients having low/negative infiltration and 3 patients having high/moderate infiltration (Figure 3D). Patients with negative CD8+ T cells in pre-treatment tissue samples were also negative in post-treatment samples. These results are in line with the reduced expression of CD8+ T cell immune genes in the IP fluid. Histologically, CD20 + B cells were identified in only two cases (pretreatment tumor tissue of one patient and recurrent tumor tissue of a second patient), and in both cases, high T cells infiltration was also observed. Both T and B lymphocyte populations were seen as diffusely infiltrating the tumor tissue or as lymphoid aggregates with distinct T and B cell clusters, similar in appearance to tertiary lymphoid structures (TLS) (Figure 3E). In line with this, TLS specific genes, such as *CCL19, CCL21,* and *CXCL13* were additionally found to be DE and upregulated in IP fluid from early to late time points [20].

### 3.4. Markers of Inflammation Differ in Systemic Circulation versus Loco-Regional Environment

We used PBMCs to explore differences in immune gene expression between the TME and peripheral immune cells. 5 patients had available paired pre- and post-treatment PBMCs, and 51 genes were DE (Appendix A) with 95% being upregulated similar to IP fluid and tumor tissue (Figure 4A). Comparison of all three sample types (fluid, tumor tissue and PBMCs) identified only 2 DE genes—*CXCR4* and *IFIT1*—which do not show concordant expression between samples (Appendix A). Individual comparisons between PBMCs and IP fluid or tumor tissue also show minimal overlap in DE genes with concordant expression (Figure 4B).

### 3.5. Secondary Patient Cohort Shows Similar Immune Gene Upregulation and B Cell Pathway Activation in Response to Chemotherapy

To validate our findings of B cell upregulation early in IP chemotherapy, we used a previously established patient cohort treated on a Phase I clinical trial with IP cisplatin, followed by IP chemokine modulation (NCT02432378). Patient characteristics included all recurrent EOC with mostly high-grade serous histology and on 2nd line chemotherapy [15]. To detect genes triggered solely by IP cisplatin, we used IP fluid samples collected at baseline and after the first treatment cycle, which consisted of chemotherapy alone. A total of 137 genes were DE in IP fluid after cycle 1 compared to baseline (Appendix A) and 98% had upregulated expression (Figure 5A), mirroring the upregulation seen above (Figure 2A). 8 genes were DE in both cohorts and of these four are specific to B cell function all of which showed concordant expression (*PAX5, POU2AF1, CD79B* and *BLK*; Figure 5B,C). IPA^®^ resulted in two B-cell specific pathways in the top 10 canonical pathways identified (Appendix A) which are also found in the IP fluid and tumor tissue of the main patient cohort. These results point to concordance of B cell involvement early in chemotherapy.

### 3.6. Anti-MUC1 Antibodies Are Present in the IP Fluid

The presence of tumor specific antibodies in the IP fluid of our original patient cohort was investigated as antibody production is a known function of B cells. We focused on antibodies to MUC1, an oncoprotein that is differentially glycosylated in cancer cells and is overexpressed in >90% of all EOC [21]. MUC1 IgM was detected in IP fluid of at least 1 time point in 5 out of 8 patients (Figure 6A). We were unable to identify trends in the presence of antibody over time, due to our limited sample size. ELISA of MUC1 IgG showed no positivity in this cohort. MUC1 IgM was also measured in plasma at corresponding time points, one early plasma time point was unavailable. Out of 20 total time points, 7 were incongruent when comparing antibody positivity in plasma versus IP fluid. In all instances of incongruency, antibody was found to be positive in plasma but negative in IP fluid (Figure 6B). These findings reveal loco-regional presence of antibodies specific to a tumor-associated antigen and are in line with gene expression data pointing to activation of Th2 and B cell function.

## 4. Discussion

We have characterized the EOC immune TME at several timepoints during IP platinum-based chemotherapy showing that the immune TME is actively changing, as evidenced by the large number of immune genes that are DE between chemotherapy time points, in both IP fluid and tumor tissue. Interestingly, most genes are upregulated pointing to an active immune microenvironment even at the end of cytotoxic chemotherapy. We also found a concentration of B cell specific genes and pathways that are upregulated, suggesting an important role of B cells in EOC tumor involution. Tumor specific antibodies were identified in IP fluid in over half of our cohort, pointing to a potential role of local B/plasma cells and their secreted antibodies in tumor involution. Simultaneously, we highlight the upregulation of Th2 immune genes and concomitant downregulation of Th1 immune genes, supporting the view that immune interventions (with vaccines or immune modulatory agents) that augment type 1 immunity are needed during or post chemotherapy [15,22].

The value of this study lies in the data from serial collection of biological samples. Our cohort included only patients with EOC treated with IP platinum-based chemotherapy, which allowed for the collection of serial IP fluid. The inclusion of IP fluid analysis provides an assessment of dynamic changes in the TME during tumor involution and a more complete picture of the TME when compared to surrogate markers in peripheral blood or static points of evaluation such as tumor tissue. The small numbers of DE genes shared between PBMCs and either IP fluid or tumor tissue lend further support to IP fluid being a more accurate reflection of the TME. Differences in microRNA and gene expressions between IP fluid and peripheral blood have been previously reported and support our findings [12,23].

Evolving evidence suggests that platinum agents have immune modulatory properties [24,25]. In addition, histological analysis of ovarian cancer tumor tissue pre- and post-neoadjuvant chemotherapy (NACT) has shown that chemotherapy can upregulate TILs including CD8, CD4, and CD20 infiltration [26,27,28]. Our study provides additional data pointing to upregulation of immune function in ovarian cancer treated with chemotherapy. Although no changes in T cells were detected, we found a B cell specific gene upregulation in both of our patient cohorts, most pronounced early in chemotherapy. In support of the importance of B cell function, we found upregulation of Th2 immune genes and complement genes shared between IP fluid and tumor tissue and are the first to report the presence of anti-MUC1 antibodies in IP fluid. While no anti-MUC1 IgG were detected in either IP fluid or plasma, most patients had IgM responses. The absence of isotype switch may be due to MUC1 glycosylation and interference with antigen presentation, as we have previously established [29]. Nevertheless, IgM antibodies are particularly effective at complement activation [30], in line with several complement genes that were upregulated in both IP fluid and tumor. In our previously published work, we have mechanistically demonstrated that complement upregulation occurs in ovarian cancer (and precursor lesions) and plays a role in tumor-promoting inflammation. We also demonstrated that downregulation of complement in epithelial cells leads to decreased cell proliferation, further supporting its role in tumor progression [31]. Although the focus of our previous study was on endometriosis-associated ovarian cancer, it is likely that similar mechanisms would be at play in high grade serous ovarian cancer [32], as complement biology has been implicated in a variety of solid tumor types [33].

The B cells detected in tumor tissue were found in organized lymphoid structures surrounded by T cells, consistent with other studies, and resemble the formation of TLS [34,35]. TLS are ectopic lymphoid formations with a T cell zone and proliferating B cell germinal center. TLS have a proposed function of efficient antigen presentation in areas, such as tumors, where there is high antigen load and a need for cytotoxic T cells. Antigen presentation in TLS may represent another mechanism through which B cells contribute to tumor involution. Few studies have investigated the role of B cells and TLS in EOC; however, the available data supports the role of B cells in the antitumor response and increased patient survival [34,36,37]. Our study further highlights the role of B cells in ovarian cancer biology as shown by studies in other tumor types [35,38,39]. For example, a recent study demonstrated that formation of B cell-rich TLS structures is associated with epitope spreading and predicts the occurrence of clinical responses to DC vaccines in patients with advanced PD1-resistant melanoma [40]. Additionally, results from studies on humoral immunity in response to vaccines for influenza or COVID-19 in patients with solid tumors treated with cytotoxic therapies also reveal a preserved ability for seroconversion during or after treatment, further highlighting the conserved state of B cell responsiveness during chemotherapy [41,42]. Interestingly, the CA125 monoclonal antibody vaccine for ovarian cancer oregovomab administered during chemotherapy was more immunoreactive than when administered after tumor remission, similarly pointing to chemo-induced B cell activation [43,44].

Data from our secondary cohort showed that chemo-immunotherapy combination treatment in EOC leads to upregulation of Th1 immunity in the TME. Analysis of IP fluid after combination treatment showed increase in T cell recruitment chemokines and upregulation of IFN-stimulated genes and IFNγ response genes [15]. This contrasts with our chemotherapy-alone cohort which shows upregulation of B cell function and Th2 immunity suggesting that inclusion of immune modulation may be necessary to trigger Th1 immunity and T cell lympho-attraction, to optimize tumor response to treatment.

The limitations of our study include a small and heterogenous sample size. However, with patient matched samples we were able to directly compare changes in the immune gene expression at serial time points during treatment. Heat maps of DE gene expression of IP fluid (Figure 2A,C) and PBMCs (Figure 4A) suggest potential outliers driving the DE detection in our small sample size (P2 in Figure 2 and P5 in Figure 4). To circumvent this issue, we performed sensitivity analyses by comparing the log-fold-change (log2FC) including P2 and P5 and the log2FC without P2 and P5 in each corresponding comparison. The majority of DE genes remain consistent without the inclusion of the potential outliers (Appendix A) although for some genes, inclusion of P2 and P5 indeed has a bigger impact. Due to the small and heterogenous nature of the patient cohort, it is difficult to distinguish whether the two samples are truly outliers or biological facts. P2 and P5 (both with high grade serous late stage EOC and adequate sample preparation) were maintained in our analyses for exploratory purposes given the reported sensitivity of analyses above.

Our sample size did limit our ability to compare survival data or other clinical outcomes. Not all patients had the same time points available for biological specimens due to challenges associated with consistent IP fluid sample collection.

## 5. Conclusions

In summary, we have been able to describe longitudinal changes in immune gene expression in the ovarian TME during treatment with platinum-based IP chemotherapy. We found that along with general upregulation of immune function, B cell biology in association with increased Th2 immunity and the complement system may play an important role in chemotherapy-induced tumor involution. No changes in anti-tumor CD8 cytotoxic cell mediated mechanisms were detected, pointing to the need for Th1/CD8 boosting immune interventions. Further studies, using similar sample collection protocol and multi-omics approaches will allow us to tailor combinations of traditional chemotherapy with Th1 boosting immunotherapy (via immune modulation and DC vaccination for example) [45] and to understand their specific contributions to tumor involution when combination treatments are used.

## Figures and Tables

**Figure 1 cancers-14-05647-f001:**
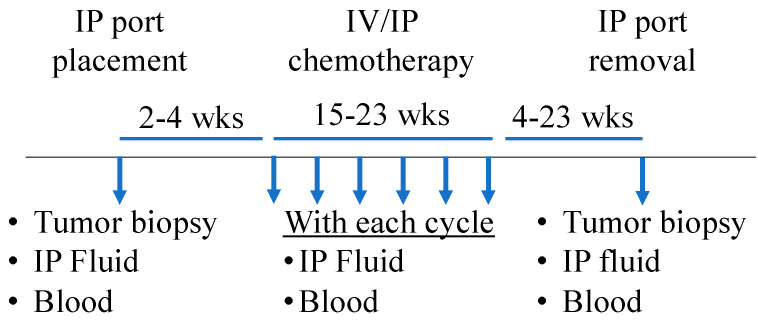
Sample collection schematic.

**Figure 2 cancers-14-05647-f002:**
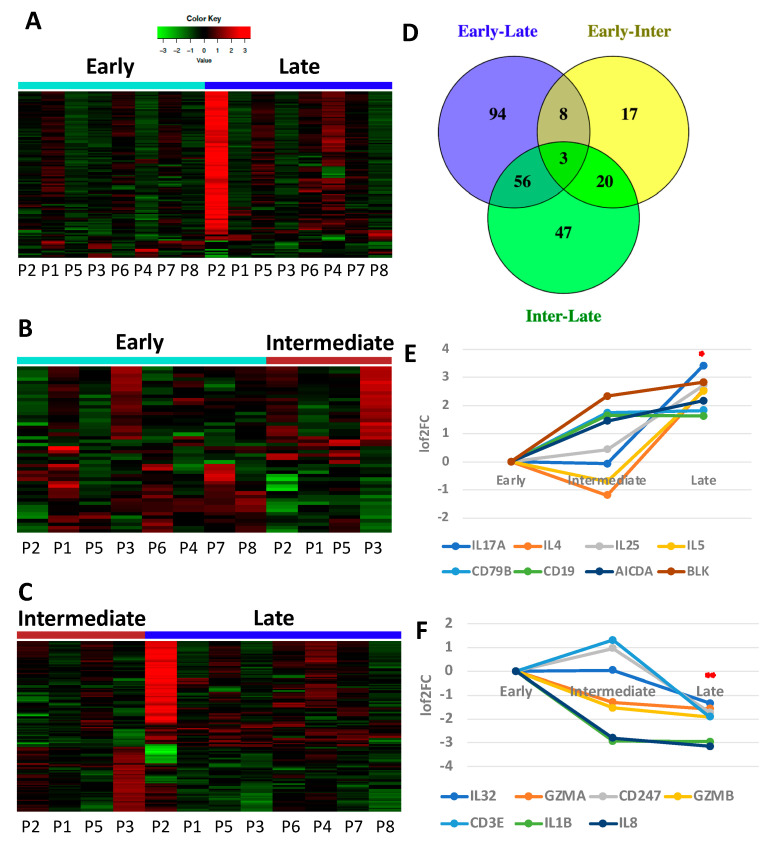
Immune gene and cytokine changes in IP fluid between early, intermediate and late time points during chemotherapy. (**A**) Heat map for DE genes (*q* < 0.2) between early and late time points shows overall upregulation of immune genes. (**B**) Heat map for DE genes (*p* < 0.05) between early and intermediate time points also shows overall upregulation early in chemotherapy. (**C**) Heat map for DE genes (*p* < 0.05) between intermediate and late time points confirms more downregulation than other time point comparisons. (**D**) Venn diagram demonstrates DE genes that are shared between time comparisons (*p* < 0.05). (**E**) Th2 immune genes (*IL17A, IL4, IL25, IL5*) and B cell specific genes (*CD79B, CD19, AICDA, BLK*) show upregulation throughout chemotherapy, * denotes significant change between early and late time points, *p* < 0.05. (**F**) Conversely Th1 immune genes show downregulation, **denotes significant change between early and late time points, *p* < 0.05 and *q* < 0.2. (**G**) Concentration of select cytokines in IP washes, via MSD, showing upregulation at intermediate and late time points. (**H**) Normalized ratios of cytokine concentrations in IP wash divided by concentrations found in plasma (the latter also detected by MSD).

**Figure 3 cancers-14-05647-f003:**
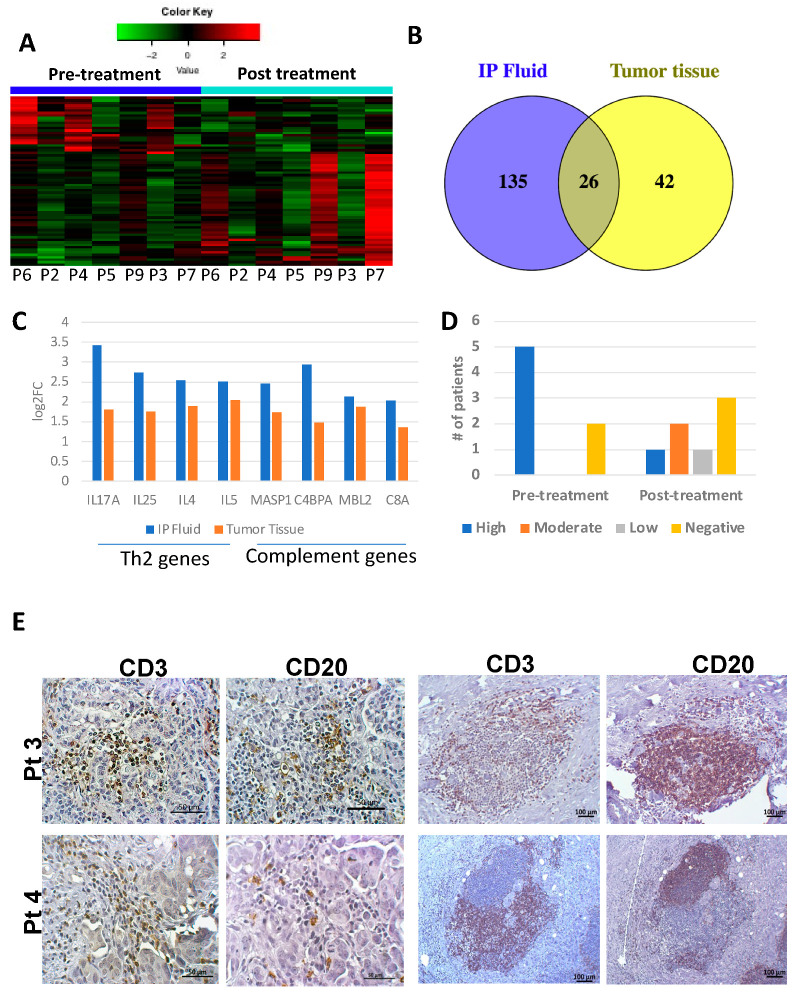
Immune gene changes between pre-treatment and post-treatment tumor tissues. (**A**) Heat map for 68 DE genes (*p* < 0.05) showing overall upregulation. (**B**) Venn diagram demonstrating that 26 genes are DE from beginning to end of chemotherapy treatment in both tumor tissue and IP fluid. (**C**) These include genes involved in Th2 immune response and the complement pathway which are significantly upregulated in both IP fluid and tumor tissue (*p* < 0.05). (**D**) IHC shows an overall decrease in tumor infiltrating CD8+ lymphocytes after chemotherapy in line with decrease in Th1 immune genes in IP fluid. (**E**) IHC staining for CD20 and CD3 in tumor tissue in two patients (Pt 3 and 4) showing tumor infiltrating B and T lymphocytes with diffuse distribution (**left panels**) or as lymphoid aggregates (**right panels**).

**Figure 4 cancers-14-05647-f004:**
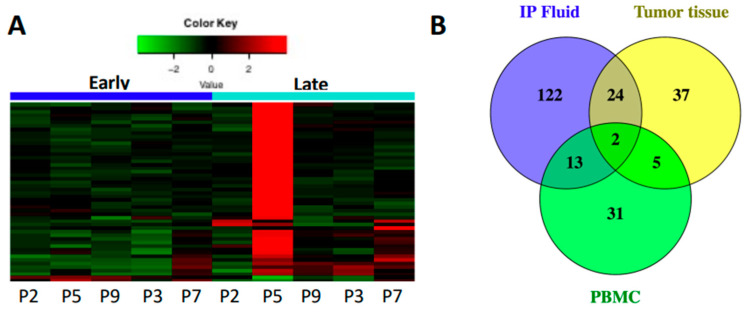
Immune gene expression of PBMC during chemotherapy. (**A**) Heat map for 51 DE genes between early and late time points (*p* < 0.05) showing overall upregulation. (**B**) Venn diagram of IP fluid, tumor tissue and PBMC DE genes show only 2 genes are DE across all three sample types, both of which have discordant expression. Additionally, few genes are shared between PBMC and either IP fluid or tumor tissue with only around half having concordant expression.

**Figure 5 cancers-14-05647-f005:**
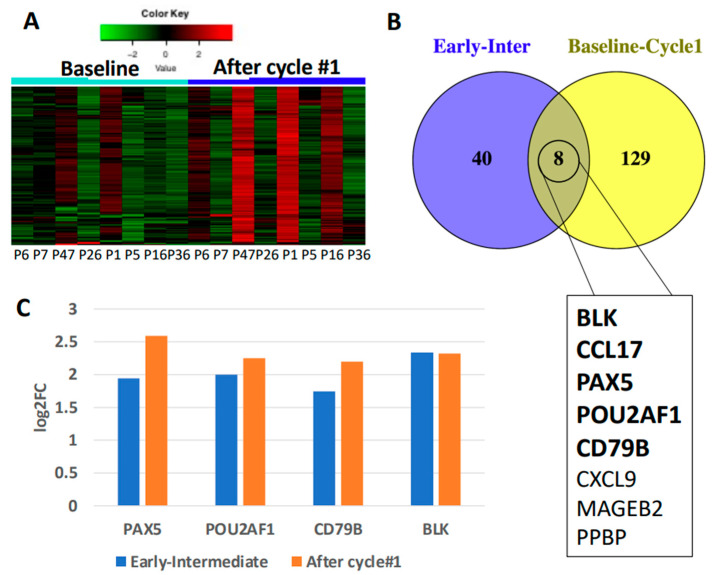
Immune gene expression analysis after 1 cycle of IP platinum-based chemotherapy in secondary patient cohort. (**A**) Heat map of 137 DE genes showing upregulation of 98% of genes. (**B**) 8 genes are DE in IP fluid from both patient cohorts. Bolded genes have concordant expression between both cohorts. (**C**) 4 out of 5 concordant genes are specific to B cell function, all of which are significantly upregulated in both cohorts (*p* < 0.05).

**Figure 6 cancers-14-05647-f006:**
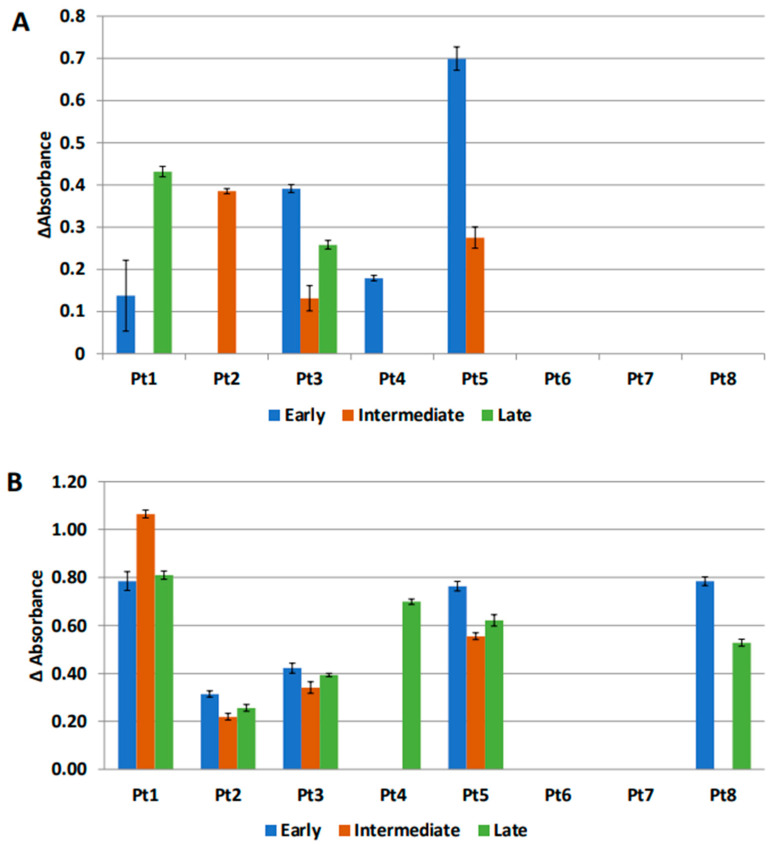
MUC1 IgM absorbance by ELISA in IP fluid (**A**) and plasma (**B**). Samples from eight patients (Pt1-Pt8, *x* axis) were collected at early (blue), intermediate (red), and late (green) time points. Values on the *y* axis were calculated as the mean difference between the experimental and background absorbance for each sample. Samples were run in triplicate and bars represent standard deviations. Negative absorbance values are denoted as zero absorbance.

**Table 1 cancers-14-05647-t001:** Patient demographics of main patient cohort.

	Patients (N = 9)
Median Age (range)	52 (25–67)
Stage	
I	1 (11%)
II	1 (11%)
III	7 (78%)
IV	0 (0%)
Grade	
1	2 (22%)
2	0 (0%)
3	7 (78%)
Histology	
Serous	7 (78%)
Endometrioid	1 (11%)
Clear Cell	1 (11%)
Line of Therapy	
1	5 (56%)
2	2 (22%)
>2	2 (22%)
Platinum Sensitive	
Yes	7 (78%)
No	2 (22%)

## Data Availability

The complete NanoString data sets are available in the Gene Expression Omnibus database under accession number GSE217179.

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
