# Peer review of "Longitudinal Modulation of Loco-Regional Immunity in Ovarian Cancer Patients Receiving Intraperitoneal Chemotherapy"

_cancers, 2022, doi:10.3390/cancers14225647_

Round 1
Reviewer 1 Report
This study presents relevant information regarding the impact of TME on chemotherapy response. Easy to read and with a specific and clear goal.
I truly believe the results obtained here are interesting in order to characterize the TME changes before and after intraperitoneal chemotherapy which is an exciting approach to managing ovarian cancer.
However, I have some questions that might be addressed before this paper is accepted:
1- Sample size: it is too low for the conclusions here stated. In the discussion , lune 353 to 354 authors mentionated " the value of this study lies in the robust data from serial collection of biological samples". Sample size is 9 and despite of sample collection at different time points, higher number of patients is needed in order to consider the results here considered as robust. Is there any chance of including more patients to this study?
2-Figure 2: These graphics represent diferences in concentrations in cytokine secretion at diferent time-points. Either IL-4 and IL-5 showed higher point desviation compared with the other time points. Please, comment why this is happening.
3- In the discussion , you have mentionated the complement system might have a role in ovarian cancer. Have you tested this hyphotesis?
Author Response
We are thankful you for the opportunity to resubmit a revised mansucript. We are also grateful to the reviewers for assessing our work and providing insightful comments and suggestions for improvement. Below are our answers, point-by-point and a description of actions taken to address all comments.
Response to Reviewer 1
1- Sample size: it is too low for the conclusions here stated. In the discussion , lune 353 to 354 authors mentionated " the value of this study lies in the robust data from serial collection of biological samples". Sample size is 9 and despite of sample collection at different time points, higher number of patients is needed in order to consider the results here considered as robust. Is there any chance of including more patients to this study?
We agree that this is a small cohort and we have removed the word “robust” from the text. In the Discussion, we acknowledge the limited patient cohorts, which are often a factor inherent to clinical studies. Unfortunately, we currently are not able to include more patients in this study. A minority of ovarian cancer patients undergo IP chemotherapy and serial sampling of the peritoneal cavity with IP washes, during aggressive chemotherapy of patients with terminal stage disease is challenging. Due to limitations in sample collection, this is the most patients we were able to include in this current study. We point out however, that despite these clear limitations, there are several advantages to our study: consistent treatment implementation by a single physician, standardized sample acquisition at a single institution and single laboratory for consistent sample processing, all of which help eliminate the variability inherently associated with larger, multi-investigator and/or multi-institutional studies.
2-Figure 2: These graphics represent differences in concentrations in cytokine secretion at different time-points. Either IL-4 and IL-5 showed higher point deviation compared with the other time points. Please, comment why this is happening.
We acknowledge the deviation. and note that it is difficult for us to pinpoint the exact cause. One possible explanation is that changes in IL4 and IL5 may follow different kinetics in different patients, reflected by the higher variability at intermediate time points. These changes may however “mature” as treatment progresses.
3- In the discussion , you have mentionated the complement system might have a role in ovarian cancer. Have you tested this hyphotesis?
We are very grateful for this question, and we have edited our manuscript (Discussion) to further clarify this statement. Indeed, in our previously published work (Suryawanshi et al, Clinical Cancer Research PMID: 25294912, https://pubmed.ncbi.nlm.nih.gov/25294912/ ) we have mechanistically demonstrated that complement upregulation occurs in ovarian cancer (and precursor lesions) and plays a role in tumor-promoting inflammation. We also demonstrated mechanistically that downregulation of complement in epithelial cancer cells leads to decreased cell proliferation, further supporting its role in tumor progression. Although the focus of our previous study was on endometriosis-associated ovarian cancer, complement biology has been implicated in a variety of solid tumor types (PMID: 28248200) and it is likely that similar mechanisms would be at play in high grade serous ovarian cancer (PMID: 34359708).

Reviewer 2 Report
This is a study in a small EOC cohort to examine immune response changes to chemotherapy via IP. While several previous studies have attempted looked at this question, this one to my knowledge is the first to examine the issue in IP setting. The results are important for the field as there has been a strong resurgence to consider IP chemo as a standard for some subset of patients. The current investigations may help shed light on the beneficiaries of IP chemo, although more work is needed.
Overall, the study is quite nice and strength/novelty clearly lies in longitudinal investigations in the IP setting.
My general feedback:
1) A lot of new work in the TLS field suggests several markers are needed to identify early vs mature TLS's. In Fig. 3, CD8 and CD19 needs to be complemented by at least some of these other markers including but not limited to BLC6, HEV, CXCL13, CC20, among others. The current serial sections make it difficult esp for pt 3 to identify the localization of the TLS cell types. These markes should be carefully re-examined with representative examples of the stain as well as all of the patient data should be enumerated in a graphical format.
2) Have any of these presented data been statistically analyzed? With the exception of Fig 6, are any of the changes statistically different? Can the authors provide these analysis as primary data as well as specifying the error bars in Fig 6?
3) Can the authors reproduce some of the key heat map genes into individual graphs? For example, Fig 2D - plot the differences in expression of the key genes across patients. What are the negative controls in this case? What are the negative controls for Fig. 6?
4) Fig 2 - would it be possible to plot the plasma levels on the same graph as the ascites and perform statistical analysis? Are there any healthy donor controls?
5) It does not appear that the Nanostring data is deposited on a public repository. If there is a reason, it cannot, this should be noted in the Data Availability Statement. Otherwise, the raw de-identified/anonymized data should be made available.
Author Response
We are thankful you for the opportunity to resubmit a revised mansucript. We are also grateful to the reviewers for assessing our work and providing insightful comments and suggestions for improvement. Below are our answers, point-by-point and a description of actions taken to address all comments.
Response to Reviewer 2
1) A lot of new work in the TLS field suggests several markers are needed to identify early vs mature TLS's. In Fig. 3, CD8 and CD19 needs to be complemented by at least some of these other markers including but not limited to BLC6, HEV, CXCL13, CC20, among others. The current serial sections make it difficult esp for pt 3 to identify the localization of the TLS cell types. These markes should be carefully re-examined with representative examples of the stain as well as all of the patient data should be enumerated in a graphical format.
We agree that research on TLS formation in solid tumors has rapidly progressed and that several criteria are now proposed to systematically characterize intratumor immune cell distribution and TLS formation (as recently reviewed, among others, by Fridman et al, Nat Rev Clin Oncol PMID: 35365796). We emphasize however that, while the tumor tissues in our cohort showed varying levels of T cells (enumerated and showed in graph format in Figure 3D), we have identified B cells in only two samples, shown in the newly revised Fig 3E. In both cases, the tumor was highly infiltrated by CD3 T cells. The distribution of T and B cells in these two cases showed up as either diffuse infiltration, lymphoid aggregates and TLS (as shown by the new CD20/CD3 stains). We have also added new details about TLS tissue localization: patient # 3 shows numerous lymphoid aggregates throughout the tissue section and 6 TLS formations, of which 3 were peritumoral and 3 were intratumoral. Patient 4 had a very high number of intratumoral TLS and lymphoid aggregates, collectively amounting to 25 % of the entire tumor tissue. Given that TLS was observed in only 2 cases, we have not extensively profiled these formations for markers such as CD21 or CD23. Consequently, we cannot ascertain the maturation status of these TLS. However, morphologically, these lymphoid structures seem to lack germinal centers, pointing to their likely immature state.
2) Have any of these presented data been statistically analyzed? With the exception of Fig 6, are any of the changes statistically different? Can the authors provide these analysis as primary data as well as specifying the error bars in Fig 6?
We appreciate the comment and have now expanded the figure legends to include additional details regarding the statistical tests performed. To reiterate, the data in 2E, 2F, 3C, 5C show changes in gene expression (log2FC). All the genes shown are significantly changed (p<-0.05). The statistical methods are described in the Methods section of the manuscript and the individual (p and adjusted q) values of these changes (interm vs early, late vs interm and late vs early) are listed in Supplemental Tables 1, 2, 3, 6 and 11.
Cytokine concentrations measured using the MSD platform were statistically analyzed and were not found to be significant, although trends were identified. The IHC data was also not statistically analyzed due to the small sample size. The values plotted in Fig. 6 represent averages and the error bars in figure 6 are the standard deviation of the absorbance values for each time point, which are run in triplicate, as detailed in the Methods section.
3) Can the authors reproduce some of the key heat map genes into individual graphs? For example, Fig 2D - plot the differences in expression of the key genes across patients. What are the negative controls in this case? What are the negative controls for Fig. 6?
Indeed, we believe that it would be beneficial to plot separately some of these genes. We have further edited the legend of figure 2 to specify that panels 2E and 2F capture changes of genes shown in the heatmaps. Average values of log2 fold changes of key genes are plotted in Figure 2E and 2F for Th2, B-cell and Th1 specific genes. As detailed in Materials and Methods, gene expression is normalized to 20 housekeeping genes which serve as negative controls.
We have further edited the Methods section to detail how the readings of each sample were corrected for “background”. Each sample was run in triplicate (serum) or duplicate (ascites) and was incubated in either against MUC1 peptide or albumin (PBS) coated wells. The plotted values represent the difference (delta) between the experimental and background OD values. Each time ELISA was run, a positive control -a patient sample known to have high antibody levels, selected from a patient cohort of an unrelated study- was also included.
4) Fig 2 - would it be possible to plot the plasma levels on the same graph as the ascites and perform statistical analysis? Are there any healthy donor controls?
We appreciate the comment, and we welcome the opportunity to explain this further. We agree that showing both plasma and IP wash measurements would be ideal. We note however, that the IP wash IL-4,-5, -6 concentrations are much higher, compared to plasma. Therefore, plotting all on the same graph would be more difficult, given the difference in scale. It is for this reason that we chose to plot (Fig 2H) the ratio of IP Wash/Plasma and discussed how cytokine and chemokine-rich IP washes reflect the highly inflammatory milieu in IP washes, including the presence of the above mentioned Th2 cytokines.
While our assays do not have “controls”, we reiterate that our goal is to perform longitudinal profiling and to have each patient serve as her own control and to compare the kinetic changes during cisplatin treatment, from early to intermediate and later time points.
5) It does not appear that the Nanostring data is deposited on a public repository. If there is a reason, it cannot, this should be noted in the Data Availability Statement. Otherwise, the raw de-identified/anonymized data should be made available.
We apologize for the omission, which we have now rectified. The complete NanoString data sets have been uploaded and are available in the Gene Expression Omnibus database under accession number GSE217179.

Round 2
Reviewer 1 Report
Thanks a lot for your answers and the changes performed in the manuscript.
I think it looks good for my side.
Best,
Author Response
Thank you for your feedback.
Reviewer 2 Report
Thank you for the responses to my original review.
New Fig 3E. While I agree that there are T and B cells visible by staining, I am not comfortable with the authors referring to these as TLS without additional markers. The quality of the image is low and it is hard to discern that they are serial sections. I would offer that the authors not use the term TLS. The small overall sample size and the small number of cases containing B cells makes it difficult to draw any inferences about TLS using only two markers.
Author Response
New Fig 3E. While I agree that there are T and B cells visible by staining, I am not comfortable with the authors referring to these as TLS without additional markers. The quality of the image is low and it is hard to discern that they are serial sections. I would offer that the authors not use the term TLS. The small overall sample size and the small number of cases containing B cells makes it difficult to draw any inferences about TLS using only two markers.
We agree and have replaced “TLS” with “lymphoid aggregates.” In line with this, we have relabeled Figure 3 and made the necessary edits throughout the text.
